# Comparison of Two Contemporary Quantitative Atherosclerotic Plaque Assessment Tools for Coronary Computed Tomography Angiography: Single-Center Analysis and Multi-Center Patient Cohort Validation

**DOI:** 10.3390/diagnostics14020154

**Published:** 2024-01-09

**Authors:** Loris Weichsel, Alexander Giesen, Florian André, Matthias Renker, Stefan Baumann, Philipp Breitbart, Meinrad Beer, Pal Maurovitch-Horvat, Bálint Szilveszter, Borbála Vattay, Sebastian J. Buss, Mohamed Marwan, Andreas A. Giannopoulos, Sebastian Kelle, Norbert Frey, Grigorios Korosoglou

**Affiliations:** 1GRN Hospital Weinheim, Cardiology, Vascular Medicine & Pneumology, 69469 Weinheim, Germany; loris.weichsel@web.de (L.W.); alexander.giesen@grn.de (A.G.); 2Cardiac Imaging Center Weinheim, Hector Foundations, 69469 Weinheim, Germany; 3Department of Cardiology, Angiology and Pneumology, University of Heidelberg, 69120 Heidelberg, Germany; florian.andre@med.uni-heidelberg.de (F.A.); norbert.frey@med.uni-heidelberg.de (N.F.); 4DZHK (German Centre for Cardiovascular Research), Partner Site Heidelberg/Mannheim, 69120 Heidelberg, Germany; 5Department of Cardiology, Campus Kerckhoff, Justus Liebig University Giessen, 61231 Bad Nauheim, Germany; m.renker@kerckhoff-klinik.de; 6DZHK (German Centre for Cardiovascular Research), Partner Site Rhein Main, 61231 Bad Nauheim, Germany; 7Department of Cardiology, District Hospital Bergstraße, 64646 Heppenheim, Germany; stefan.baumann@kkh-bergstrasse.de; 8First Department of Medicine-Cardiology, University Medical Center Mannheim, 68167 Mannheim, Germany; 9Department of Cardiology and Angiology, Medical Center-University of Freiburg, Faculty of Medicine, University of Freiburg, 79189 Bad Krozingen, Germany; philipp.breitbart@uniklinik-freiburg.de; 10Department for Diagnostic and Interventional Radiology, University Hospital Ulm, 89081 Ulm, Germany; meinrad.beer@uniklinik-ulm.de; 11Heart and Vascular Center, Semmelweis University, 1122 Budapest, Hungary; maurovich.horvat@gmail.com (P.M.-H.); szilveszter.balint@gmail.com (B.S.); bori.vattay@gmail.com (B.V.); 12MVZ-DRZ Heidelberg, 69126 Heidelberg, Germany; prof.buss@mvz-drz.de; 13Department of Cardiology, University of Erlangen, 91054 Erlangen, Germany; mohamed.marwan@uk-erlangen.de; 14Department of Nuclear Medicine, Cardiac Imaging, University Hospital Zurich, 8091 Zurich, Switzerland; andreas.giannopoulos@usz.ch; 15Deutsches Herzzentrum der Charité, Department of Cardiology, Angiology and Intensive Care Medicine, Charité-University Medicine Berlin, 10117 Berlin, Germany; sebastian.kelle@dhzc-charite.de

**Keywords:** plaque quantification, non-calcified, calcified, plaque volume, composition, serial studies, inter- and intra-observer variabilities

## Abstract

Background: Coronary computed tomography angiography (CCTA) provides non-invasive quantitative assessments of plaque burden and composition. The quantitative assessment of plaque components requires the use of analysis software that provides reproducible semi-automated plaque detection and analysis. However, commercially available plaque analysis software can vary widely in the degree of automation, resulting in differences in terms of reproducibility and time spent. Aim: To compare the reproducibility and time spent of two CCTA analysis software tools using different algorithms for the quantitative assessment of coronary plaque volumes and composition in two independent patient cohorts. Methods: The study population included 100 patients from two different cohorts: 50 patients from a single-center (Siemens Healthineers, SOMATOM Force (DSCT)) and another 50 patients from a multi-center study (5 different > 64 slice CT scanner types). Quantitative measurements of total calcified and non-calcified plaque volume of the right coronary artery (RCA), left anterior descending (LAD), and left circumflex coronary artery (LCX) were performed on a total of 300 coronaries by two independent readers, using two different CCTA analysis software tools (Tool #1: Siemens Healthineers, syngo.via Frontier CT Coronary Plaque Analysis and Tool #2: Siemens Healthineers, successor CT Coronary Plaque Analysis prototype). In addition, the total time spent for the analysis was recorded with both programs. Results: The patients in cohorts 1 and 2 were 62.8 ± 10.2 and 70.9 ± 11.7 years old, respectively, 10 (20.0%) and 35 (70.0%) were female and 34 (68.0%) and 20 (40.0%), respectively, had hyperlipidemia. In Cohort #1, the inter- and intra-observer variabilities for the assessment of plaque volumes per patient for Tool #1 versus Tool #2 were 22.8%, 22.0%, and 26.0% versus 2.3%, 3.9%, and 2.5% and 19.7%, 21.4%, and 22.1% versus 0.2%, 0.1%, and 0.3%, respectively, for total, noncalcified, and calcified lesions (*p* < 0.001 for all between Tools #1 and 2 both for inter- and intra-observer). The inter- and intra-observer variabilities using Tool #2 remained low at 2.9%, 2.7%, and 3.0% and 3.8%, 3.7%, and 4.0%, respectively, for total, non-calcified, and calcified lesions in Cohort #2. For each dataset, the median processing time was higher for Tool #1 versus Tool #2 (459.5 s IQR = 348.0–627.0 versus 208.5 s; IQR = 198.0–216.0) (*p* < 0.001). Conclusion: The plaque analysis Tool #2 (CT-guided PCI) encompassing a higher degree of automated support required less manual editing, was more time-efficient, and showed a higher intra- and inter-observer reproducibility for the quantitative assessment of plaque volumes both in a representative single-center and in a multi-center validation cohort.

## 1. Introduction

Coronary computed tomography angiography (CCTA) is widely accepted as a clinical tool for the non-invasive diagnostic work-up and the risk stratification of patients with suspected or known coronary artery disease (CAD) and has been endorsed by current guidelines [1,2,3]. Several studies highlighted the ability of CCTA to assess coronary atherosclerotic plaque composition and to quantify each of the plaque components [4]. This is of great importance since high-risk features, inclusive of non-calcified plaque components, are potential precursors of plaque rupture, potentially causing acute coronary syndromes and cardiac death [5,6,7]. 

However, the accurate and reproducible analysis of atherosclerotic plaque, which is a prerequisite for the prognostic value of CCTA, is highly dependent on the software tool used for analysis, as manual analysis is tedious and time-consuming. In this regard, software tools for coronary plaque analysis are necessary, which may apply different algorithms for centerline extraction and segmentation of the inner and outer vessel walls. Such algorithms, however, may vary in terms of automatization of the analysis steps. Thus, while some tools predominantly rely on manual contouring of coronary plaques, others perform automated analysis with the option of post-processing manual editing by the operator [8].

Since such automatization of multiple analysis steps may lead to an increased reproducibility of the acquired values within a shorter time, the aim of our study was to assess the performance of (1) a predominantly manually based and (2) an automated plaque analysis software tool. The two tools were tested in a single-center cohort, whereas the results by the automatic tool were verified within a multi-center and multi-vendor patient cohort.

## 2. Materials and Methods

### 2.1. Study Design and Patient Population

The study included a total of 100 randomly selected patients from a single-center (*n* = 50) and a multi-center, multi-vendor cohort (*n* = 50). All patients underwent clinically indicated CCTA due to suspected or known CAD and had stable clinical symptoms. The patients with acute coronary syndromes (ACS) or undergoing CCTA for other indications such as planning structural heart procedures were excluded.

Traditional cardiovascular risk factors, including arterial hypertension, hyperlipidemia, current or prior smoking, diabetes mellitus, and family history of CAD as well as history of CAD, myocardial infraction, or prior percutaneous coronary intervention (PCI) were recorded in all patients. 

### 2.2. CCTA Protocol

CCTA was performed using >64 slice CT scanner technology in both cohorts. The CCTA in the single-center cohort was performed on a third-generation dual-source CT scanner (SOMATOM Force, Siemens Healthineers, Forchheim, Germany). This 192-slice dual-source CT system offers a Turbo Flash high-pitch spiral mode with 737 mm/s; maximum pitch 3.2) and a minimal gantry rotation time of 250 ms, resulting in a 66 ms temporal resolution for an electrocardiography (ECG)-triggered or gated dual-source scan. 

In the multicenter cohort, CCTA were performed with 5 different CT scanners from 3 manufacturers (Siemens Healthineers, Forchheim, Germany; Philips Healthcare, Best, The Netherlands; GE Healthcare, Chicago, IL, USA). All CT scanners enable a high pitch (>185 mm/s; max pitch 3.4) and minimum gantry rotation time of 300 ms and below, resulting in a temporal resolution of ≤150 ms (Appendix A).

Metoprolol (Cohort #1, *n* = 33 patients; Cohort #2, *n* = 39 patients) was titrated intravenously in individually adjusted doses for patients with heart rates greater than 65 beats per minute to achieve heart rates of around 60 bpm or lower. The acquisition protocol was selected based on patient-specific parameters, including heart rate and rhythm. Patients with regular heart rates ≤ 70 bpm underwent ECG-triggered prospective axial or turbo Flash high-pitch spiral mode acquisitions. Patients with heart rates > 80 bpm despite beta-blocker administration or with frequent ectopic beats were scanned using ECG-triggered prospective arrhythmia protocols, triggering within the systole of the cardiac cycle or spiral acquisitions with retrospectively ECG-gated image reconstruction (details provided in Appendix A).

### 2.3. Plaque Burden Quantification Using Plaque Analysis Software

Plaque analysis was performed by two different investigators (LW and AG), as well as by the same investigator at different time points with a time difference of at least 6 weeks. A coronary plaque was defined as a structure > 1 mm^2^ located within or adjacent to the coronary artery lumen. 

Plaque volumes were assessed using a dedicated but primarily manual plaque evaluation research software (Tool #1: syngo.via Frontier CT Coronary Plaque Analysis, version 5.0.2, Siemens Healthineers, Forchheim, Germany) as well as with an automated plaque evaluation research software (Tool #2: the successor CT Coronary Plaque Analysis prototype, version V30, Siemens Healthineers, Forchheim, Germany). 

Tool #1 first carries out a fully automatic isolation of the heart and centerline detection, which can be manually corrected if necessary. After manual selection of a vessel section of interest, the tool provides an automatically generated suggestion for the inner and outer wall segmentation, which must be reviewed and manually refined where necessary. Within this obtained region of interest between the inner and outer wall, a plaque composition analysis can be carried out according to different Hounsfield Unit-based thresholds. The algorithmic details of these steps have been described by Denzinger et al. [9]. 

Tool #2, on the other hand, automatically segments coronary centerlines and lumen and vessel wall contours using deep learning algorithms. By applying the same Hounsfield Units-based thresholding as Tool #1, plaque volumes for the complete coronary tree are determined. The user can then arbitrarily modify centerlines and lumen/wall contours to refine the segmentation, which also automatically updates the resultant plaque volumes. Both software tools used the following cut-off values for the differentiation between calcified (>350 Hounsfield Units (HU)) and non-calcified (<350 HU) plaques, including fibrous (30–350 HU) and lipid (<30 HU) plaque components with corresponding CT values [10].

To quantify coronary plaque components, the plaque burden of all three coronary vessels (right coronary artery (RCA), left anterior descending artery (LAD), and left circumflex artery (LCX)) and the total plaque burden were determined for each patient, differentiating between calcified and non-calcified plaques. The automated segmentation of both software tools was reviewed and corrected by the observers. 

### 2.4. Time Spent and Image Quality

The time required for the analysis per case, including the time from loading the CCTA dataset to the results of plaque analysis, was measured.

In addition, the CCTA datasets were evaluated for image quality based on a 5-point grading scale system as described previously [11]. Score 1 denotes excellent image quality (without artifacts); 2 denotes very good image quality (sharp vessel margins, minor artifacts); 3 denotes good image quality (blurring of vessel margin and minor artifacts); 4 denotes adequate image quality (notably blurred vessel, but acceptable for diagnosis); and 5 denotes non-diagnostic image quality (severe artifacts with insufficient delineation between lumen and surrounding tissues).

### 2.5. Statistical Analysis

The analyses were performed using MedCalc for Windows, version 20.009 (MedCalc Software, Ostend, Belgium). Normal distribution was tested using the Shapiro–Wilk test. All continuous variables were non-normally distributed and are therefore reported as median with interquartile ranges (IQRs). Categorical variables were reported as numbers and proportions. Categorical data were compared using x^2^ tests or the Fischer’s exact test. The non-parametric Mann–Whitney U-test was used for the comparison of independent non-normally distributed continuous variables (demographic parameters). The Wilcoxon test was used for comparison of non-normally distributed paired continuous variables (time spent for analysis). Inter-observer variabilities were assessed by repeated analysis of total, calcified, and non-calcified plaque volumes in all datasets. For correlation analysis, Pearson correlation coefficients were calculated and reported, including 95% confidence intervals (CIs). Differences were considered statistically significant at *p* < 0.05. Bland–Altman plots were used to assess the limits of agreement by the two software tools for the assessment of plaque volumes.

## 3. Results

### 3.1. Demographic Data, Cardiac Medications, and CCTA Characteristics

The patients in Cohort #1 were 62.0 (55.0–70.0) years old, 10 (20.0%) were female, and 4 (8.0%) had diabetes mellitus. The patients in Cohort #2 were 69.5 (64.0–80.0) years old, 35 (70.0%) were female, and 7 (14.0%) had diabetes mellitus (Table 1). 

### 3.2. Image Quality

The image quality was similar between Cohorts #1 and #2 with a mean score of 2.0 (1.8–2.5) versus 2.3 (1.75–2.81) (*p* = 0.25). 

### 3.3. Inter- and Intra-Observer Variabilities for Total Plaque Volumes in All Three Coronary Vessels

In Cohort #1, inter-and intra-observer variabilities for Tool #1 versus #2 were 22.8%, 22.0%, and 26.0% and 19.7%, 21.4%, and 22.1% versus 2.3%, 3.9%, and 2.5% and 0.2%, 0.1%, and 0.3% for total, noncalcified, and calcified lesions, respectively (*p* < 0.001 for all). The overall inter- and intra-observer variabilities for Tool #1 versus Tool #2 were 23.6% and 21.1% versus 2.9% and 0.2%, respectively, in Cohort #1 (*p* < 0.001) (Table 2).

The inter- and intra-observer variabilities using Tool #2 remained low at 2.9%, 2.7%, and 3.0% and 3.8%, 3.7%, and 4.0%, respectively, for total, non-calcified, and calcified lesions in Cohort #2 (Table 2).

The correlation coefficients exhibited significant and higher reproducibility between observer 1 and observer 2 for plaque volumes using Tool #2 versus Tool #1 in Cohort #1 (r = 0.70, 95%CI = 0.61–0.77 for Tool #1 versus r = 0.99, 95%CI = 0.98–0.99 for Tool #2, Figure 1A,B, *p* < 0.001). Similarly high correlation coefficients were measured with Tool #2 in Cohort #2 (r = 0.99, 95%CI = 0.98–0.99, Figure 1C, *p* = NS versus Cohort #1). Accordingly, lower limits of agreement were noticed with Tool #2 compared to Tool #1 (Figure 1D versus Figure 1E,F). Similar correlation coefficients and limits of agreement were noted in the evaluation of plaque volumes by the same observer (values can be depicted in Figure 2A–F). The correlation coefficients were significantly higher with Tool #2 versus Tool #1, both for inter- and intra-observer variabilities (*p* < 0.001). 

### 3.4. Association of Increasing Automation with Lower Time Spent

For each dataset, the median processing time was higher for Tool #1 versus Tool #2 (459.5 s; (348.0–627.0) versus 208.5 s; (198.0–216.0)) (*p* < 0.001).

## 4. Discussion

The present study shows that plaque analysis software with a higher degree of automation and less need for manual processing for plaque analyses results in a higher reproducibility for the quantitative analysis of plaque volumes in a significantly shorter time.

Reproducible measurements of plaque burden and reliable plaque characterization are essential for risk stratification of CAD patients, particularly those undergoing CT follow-up to monitor pharmacological interventions. Thus, reproducible quantification of plaque volumes especially within serial studies is essential for judging the effectiveness and the net clinical benefit of pharmacologic strategies. 

Several studies previously demonstrated that CCTA allows for the quantification of both total, calcified, and non-calcified plaque volume with good inter-observer reproducibility [12]. Of note, the reproducibility of plaque quantification is dependent on both the readers and the plaque analysis software. Some studies have previously investigated the reproducibility of plaque quantification using various plaque analysis tools (Table 3). Thus, Gitsoudis et al., Korosogolou et al. and Lee et al. reported favorable inter-observer agreement with low limits of agreement (LOAs), demonstrating low intra-observer and inter-observer variabilities of 9% and 13%, respectively [4,13,14]. Other studies indicated even lower intra-observer and inter-observer variabilities, with values between 1.3% and 3.3% [15,16]. Meah et al. extended these findings by revealing not only minimal inter- and intra-observer variabilities but also very robust inter-scan reproducibility for the assessment of total volume and non-calcified plaque volume in 20 patients who underwent repeated CCTA within 2 weeks. Similar findings were reported by Symons et al. [17] who demonstrated good scan–rescan reproducibility with the same CCTA instrument vendor, whereas differences between CT vendors resulted in a lower scan–rescan reproducibility. While these data reinforce the reproducibility of the CCTA methodology for plaque volume quantification, radiation safety issues need to be considered with repeated CCTA scans within short time periods [18]. Therefore, inter-scan variability was not within the scope of the present study. In addition, Tzolos et al. [19] also previously demonstrated high intraclass correlation coefficients for the quantification of the non-calcified plaque burden in patients with suspected coronary artery disease, confirming the ability of CCTA as a robust method for the assessment of cardiovascular risk. This agrees with previous reports by Papadopoulou et al. [16] who demonstrated highly reproducible plaque quantification assessment by CCTA using semiautomatic software tools, which translates to the ability to use CCTA for the longitudinal assessment of plaque changes over time during specific lipid-lowing treatment regimes, as is currently being investigated within the multi-center LOCATE registry (https://www.drks.de/DRKS00031954 accessed on 12 June 2023) [20]. Our results largely agree with these previous observations. Thus, both software tools used in our study (Tool #1 and Tool #2) exhibited reproducible quantification of plaque volume. However, significant differences were found between Tool #1, which is predominantly based on manual contouring of plaques, and Tool #2, which primarily performs automated analyses with the option of manual post-processing by the reader. Thus, levels of inter- and intra-observer variability and correlation coefficients exhibited significantly higher variations for Tool #1 compared to Tool #2, which agrees with previous observations [21]. This is of particular importance in cases with less experienced operators, where manual contouring may be less reproducible with less experienced operators and can therefore be improved by an automatic analysis [21]. In addition, previous studies pointed to the need for the manual readjustment of plaque contours when using automatic software tools, which may otherwise have led to a deterioration of observer agreements [19,22]. Although the degree of automatization required for accurate and time-effective plaque quantification is possibly largely dependent on the experience of the operator, such limitations may be related to the absence of predefined rules for contour adjustments, and could be circumvented using artificial intelligence and deep learning algorithms in future studies. On the other hand, several studies have demonstrated that training and adherence to predefined rules can significantly enhance the interobserver agreement for data analyses based on contour tracking [23,24]. However, such a need for manual correction and readjustment of contours is associated with increased time, as with Tool #1 versus #2 in our study, which is also consistent with previous observations using semi-automated software tools [4].

Furthermore, the limits of agreement (LOAs) using Tool #2 were ~10-fold lower for both inter-observer and intra-observer variability compared to using Tool #1, providing reassurance that the automated plaque analysis by Tool #2 will be a reliable tool for assessing plaque volume and composition in future serial CCTA studies during pharmacological interventions, as is currently being performed in the multi-center LOCATE study (https://www.drks.de/DRKS00031954 accessed on 12 June 2023). Accordingly, previous studies noted that plaque volume assessment with CCTA correlates closely with the reference standard of intravascular ultrasound [25]. Since the latter is invasive and bears the risk of complications of an invasive procedure, CCTA may serve as a valuable tool for the serial non-invasive assessment of coronary atherosclerotic plaques in future studies. Notably, the results from Cohort #1, including patients scanned with a third-generation dual source CT (DSCT), were applicable to the real-world Cohort #2, which included CCTA scans from different centers and vendors. Despite this inhomogeneity in terms of scanner types, centers, and CCTA protocols, the high reproducibility of calcified, non-calcified, and total plaque volumes in Cohort #2 using the automated CT-guided PCI Tool #2 is encouraging.

## 5. Conclusions

Plaque burden measurements using the automated CT-guided PCI Tool #2 significantly improved the reproducibility for non-calcified, calcified, and total plaque volumes in less time compared to the standard Tool #1. The results were obtained in a single-center cohort, but were verified in a real-world, multi-center, multi-vendor dataset, where the CT-guided PCI Tool #2 demonstrated a similar low observer variability and narrow limits of agreement for coronary plaque volume quantification analysis. 

### Study Limitations

Our study has some limitations. Firstly, the number of patients included both in Cohorts #1 and 2 was relatively small. In addition, clinical characteristics, including age, gender, clinical presentation, and cardiac medications, of the patients differed between Cohorts #1 and 2, which may be attributed to local differences in terms of clinical indication for a CCTA examination between cardiac centers; these are all parameters which were not controlled for or dictated by the study protocol. Furthermore, the measures for intra- and inter-observer variabilities were obtained in the same CCTA scans only. However, radiation safety issues prohibited in vivo inter-scan variability assessments. Finally, we focused on plaque volumes per vessel and per patient, rather than an analysis per plaque, which may have been related to larger variabilities for single measurements. However, focusing on a per patient analysis is clinically more meaningful. It should also be noted that a gold standard for comparisons of the obtained plaque volumes, such as intravascular ultrasound, was not available in these cohorts.

## Figures and Tables

**Figure 1 diagnostics-14-00154-f001:**
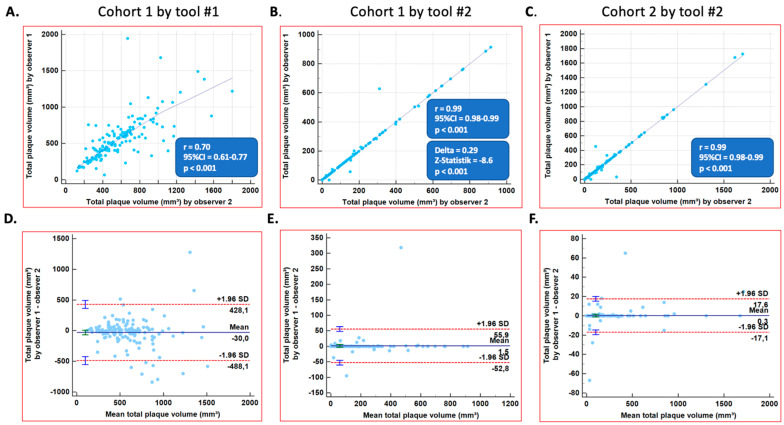
Inter-observer agreements for total plaque volumes in all 3 coronary vessels. (**A**) Total plaque volumes as assessed by tool #1 in cohort 1 (inter-observer). (**B**) Total plaque volumes as assessed by tool #2 in cohort 1 (inter-observer). (**C**) Total plaque volumes as assessed by tool #2 in cohort 2 (inter-observer). (**D**) Bland-Altman plot for total plaque volumes, as assessed by tool #1 in cohort 1 (inter-observer). (**E**) Bland-Altman plot for total plaque volumes, as assessed by tool #2 in cohort 1 (inter-observer). (**F**) Bland-Altman plot for total plaque volumes, as assessed by tool #2 in cohort 2 (inter-observer).

**Figure 2 diagnostics-14-00154-f002:**
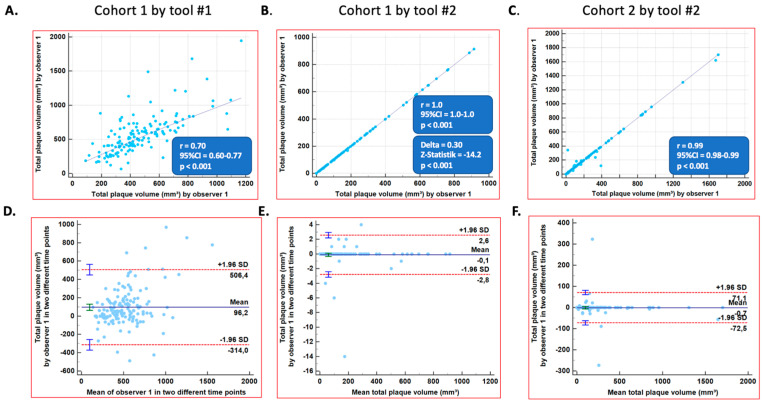
Intra-observer agreements for total plaque volumes in all 3 coronary vessels. (**A**) Total plaque volumes as assessed by tool #1 in cohort 1 (intra-observer). (**B**) Total plaque volumes as assessed by tool #2 in cohort 1 (intra-observer). (**C**) Total plaque volumes as assessed by tool #2 in cohort 2 (intra-observer). (**D**) Bland-Altman plot for total plaque volumes, as assessed by tool #1 in cohort 1 (intra-observer). (**E**) Bland-Altman plot for total plaque volumes, as assessed by tool #2 in cohort 1 (intra-observer). (**F**) Bland-Altman plot for total plaque volumes, as assessed by tool #2 in cohort 2 (intra-observer).

**Table 1 diagnostics-14-00154-t001:** Demographic, clinical data, and cardiac medications.

	Single-Center Cohort #1, *N* = 50	Multi-Center Cohort #2, *N* = 50	*p*-Values
**Baseline data and risk factors**			
Age (yrs.)	62.0 (IQR = 55.0–70.0)	69.5 (IQR = 64.0–80.0)	<0.001
Age > 60 yrs.	28 (56.0%)	40 (80.0%)	0.01
Female gender	10 (20.0%)	35 (70.0%)	<0.001
Body mass index (kg/m^2^)	28.3 (IQR = 24.5–30.3)	25.7 (IQR = 23.0–30.5)	0.04
Arterial hypertension	31 (62.0%) *	32 (64.0%)	0.83
Hyperlipidemia	34 (68.0%) **	20 (40.0%)	0.005
Diabetes mellitus	4 (8.0%) *	7 (14.0%)	0.34
Active or former smoking	13 (26.0%) *	8 (16.0%)	0.22
Family history of CAD Number of cardiovascular (CV) risk factors (0–6)	21 (42.0%) *	19 (38.0%)	0.68
	3.0 (IQR = 2.0–4.0)	2.0 (IQR = 2.0–3.0)	0.53
**History of CAD, PCI, and arrhythmias**			
Prior cardiac catheterization	20 (40.0%)	8 (16.0%)	0.008
Prior PCI	16 (32.0%)	5 (10.0%)	0.007
Prior myocardial infarction	8 (16.0%)	4 (8.0%)	0.22
Atrial fibrillation	7 (14.0%)	10 (20.0%)	0.43
**Baseline clinical presentation**			
Stable chest pain syndrome	26 (52.0%)	26 (52.0%)	1.0
Exertional dyspnea	16 (32.0%)	27 (54.0%)	0.03
Palpitations or other unspecific symptoms	8 (16.0%)	4 (8.0%)	1.0
Syncope	0 (0.0%)	1 (2.0%)	0.32
**Baseline cardiac medications**			
Aspirin	27 (54.0%)	18 (36.0%)	0.07
ß-blockers	22 (44.0%)	21 (42.0%)	0.84
Calcium antagonists	10 (20.0%)	9 (18.0%)	0.80
Diuretics	11 (22.0%)	11 (22.0%)	1.0
ACE inhibitors or AT2 blockers	13 (26.0%)	25 (50.0%)	0.01
PCSK9 inhibitors	9 (18.0%)	0 (0.0%)	0.002
Statins (independent of intensity)	35 (70.0%)	22 (44.0%)	0.009
Ezetimibe	5 (10.0%)	3 (6.0%)	0.46

* variable not assessed in 2 of 50 patients (4.0%). ** variable not assessed in 1 of 50 patients (2.0%). CAD, coronary artery disease; PCI, percutaneous coronary intervention; ACE, angiotensin-converting-enzyme; AT, angiotensin; PCSK9, proprotein convertase subtilisin/kexin type 9.

**Table 2 diagnostics-14-00154-t002:** Inter- and intra-observer variabilities (%) for Tool #1 versus #2 for total, calcified, and non-calcified plaque volumes in 3 different coronary territories and per patient.

	RCA	LAD	LCX	All 3 Coronary Arteries (per Patient)
	Cohort #1; Inter-observer variability using Tool #1 (%)			
Total plaque volume (mm^3^)	19.46	21.34	27.62	22.81
Calcified plaque volume (mm^3^)	19.30	25.06	33.63	26.00
Non-calcified plaque volume (mm^3^)	24.41	18.99	22.70	22.03
	Cohort #1; Inter-observer variability using Tool #2 (%)			
Total plaque volume (mm^3^)	1.79	0.07	4.98	2.28
Calcified plaque volume (mm^3^)	2.47	0.04	5.06	2.52
Non-calcified plaque volume (mm^3^)	1.43	5.17	4.98	3.86
	Cohort #1; Intra-observer variability using Tool #1 (%)			
Total plaque volume (mm^3^)	18.72	15.38	25.10	19.73
Calcified plaque volume (mm^3^)	17.81	19.07	29.48	22.12
Non-calcified plaque volume (mm^3^)	21.70	19.89	22.73	21.44
	Cohort #1; Intra-observer variability using Tool #2 (%)			
Total plaque volume (mm^3^)	0.27	0.00	0.28	0.18
Calcified plaque volume (mm^3^)	0.39	0.12	0.51	0.34
Non-calcified plaque volume (mm^3^)	0.07	0.07	0.10	0.08
	Cohort #2 (validation); Inter-observer variability using Tool #2 (%)			
Total plaque volume (mm^3^)	1.83	0.34	6.58	2.92
Calcified plaque volume (mm^3^)	1.77	0.24	6.84	2.95
Non-calcified plaque volume (mm^3^)	1.67	0.37	6.12	2.72
	Cohort #2 (validation); Intra-observer variability using Tool #2 (%)			
Total plaque volume (mm^3^)	6.46	1.20	3.79	3.82
Calcified plaque volume (mm^3^)	6.77	0.86	4.48	4.04
Non-calcified plaque volume (mm^3^)	5.46	1.99	3.76	3.74

**Table 3 diagnostics-14-00154-t003:** Overview of studies investigating the reproducibility of plaque analysis tools.

	Number of Patients	Number of Vessels	Scanner Type	Plaque Analysis Software	Intra-Observer Reproducibility	Inter-Observer Reproducibility
Symons et al. [17]	80	667	320-Detektor-Zeilenscanner (Aquilion One Vision; Toshiba, Otawara, Japan)	QAngioCT, version 2.1.9.1 (Medis Medical Imaging Systems, Leiden, The Netherlands)	ICC: 0.96	NA
Gitsioudis et al. [4]	521	7690	256-detector row CT scanner (iCT; Philips Medical Systems, Best, The Netherlands)	Extended Brilliance Workspace 4.0 (Philips Medical Systems)	Variability: 9%	Variability: 13% LAO: 93% (k = 0.85)
Tzolos et al. [19]	50	157	320-multidetector row scanners	Autoplaque 2.5 (Cedars-Sinai Medical Center)	ICC: 0.978 LAO ± 5.97%	ICC: 0.944 LOA ± 9.61%
Laqmani et al. [22]	10	30	256-MDCT scanner (Brilliance iCT, Philips, Best, The Netherlands)	Comprehensive Cardiac Analysis, Extended Brilliance Workspace, V4.0 (Philips Healthcare, Best, The Netherlands)	NA	LOA: −3.3 ± 33.8%
Papadopoulou et al. [16]	10	21	64-slice dual source CT scanner (Somatom Definition, Siemens Medical Solutions, Forchheim, Germany)	QAngioCT Research Edition v1.3.61 (Medis Medical Imaging Systems, Leiden, The Netherlands)	Variability: 1.30 ± 1.09%	Variability: 1.6%
Lee et al. [14]	39	15	Brilliance iCT (Philips Medical Systems, Best, The Netherlands)	Extended Brilliance Workspace V4.0; (Philips Healthcare, Cleveland, OH, USA)	LOA: −21.6 and 13.2 mm^3^	LOA: 24.6 and 20.3 mm^3^
Klass et al. [15]	35	105	Brilliance iCT (Philips Healthcare, Cleveland, OH, USA)	Cardiac Viewer and Comprehensive Cardiac Analysis, Brilliance Workspace (Philips Healthcare, Cleveland, OH, USA)	NA	Variability: 3.3%
Korosoglou et al. [13]	27	81	256-slice Brilliance iCT (Philips Medical Systems)	Plaque SW version 4.0.2 (Extended Brilliance Workspace 4.0, Philips Medical Systems)	Variability: 9%	Variability: 13%
Meah et al. [18]	20	NA	64-multidetector Biograph mCT (Siemens Medical Systems, Erlangen, Germany)	AutoPlaque, Version 2.5, Cedars-Sinai Medical Center, Los Angeles, CA, USA	Variability: 2.6% LCCC: 1.0 (1.0–1.0)	Variability: 13.5% LCCC: 0.97 (0.93–0.99)

NA, not applicable; LOA, limits of agreement; ICC, intraclass correlation coefficient; LCCC, Lin’s concordance correlation coefficient.

## Data Availability

The data presented in this study are available on request from the corresponding author.

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
