# Peer review of "Comparison of Two Contemporary Quantitative Atherosclerotic Plaque Assessment Tools for Coronary Computed Tomography Angiography: Single-Center Analysis and Multi-Center Patient Cohort Validation"

_diagnostics, 2024, doi:10.3390/diagnostics14020154_

Round 1
Reviewer 1 Report
Comments and Suggestions for Authors
Line 40-41 RCA,LAD, LCX abbreviations are not explained
Line 44 information on the age and sex of participants is more likely for materials and methods part of abstract; about diabetes – is not necessary in the abstract - why not about dyslipidemia etc
Line 101 ECG
Lines 138-140 written as only tool 2 differentiates calcified and non-calcified plaques
Line 258 LOA
Line 268 DSCT
Question: in table 3 all references except Tzolos et al are rather old and all used another software not siemens. Do you know did somebody else investigated reproducibility of siemens software and maybe compared siemens software with software listed by you in table 3?
Author Response
Thank you very much for taking the time to review this manuscript. Please find the detailed responses below and the corresponding revisions/corrections highlighted/in track changes in the re-submitted files.
|
|
Reviewer 1 |
Reply |
|
Line 40-41 |
RCA, LAD, LCX abbreviations are not explained |
We thank you for the suggestion. The Abbreviations have now been inserted as suggested. |
|
Line 44 |
information on the age and sex of participants is more likely for materials and methods part of abstract; about diabetes – is not necessary in the abstract - why not about dyslipidemia etc. |
We thank the reviewer for the suggestion. We now indeed deleted the information about the presence of diabetes from the abstract, as suggested and added information on dyslipidemia instead. Since this information is part of the Results, we decided to keep this information in this section of the abstract. |
|
Line 101 |
ECG |
We thank you for the suggestion. The Abbreviation has now been inserted as suggested. |
|
Lines 138-140 |
written as only tool 2 differentiates calcified and non-calcified plaques |
We thank you for your comment. Line 138 already includes "both software tools..." |
|
Line 258 |
LOA
|
We thank you for the suggestion. The Abbreviation has now been inserted as suggested. |
|
Line 268 |
DSCT |
We thank you for the suggestion. The Abbreviation has now been inserted as suggested. |
|
Question |
in table 3 all references except Tzolos et al are rather old and all used another software not siemens. Do you know did somebody else investigated reproducibility of siemens software and maybe compared siemens software with software listed by you in table 3? |
We thank you for the suggestion. We now inserted a study from a more recent study, as suggested by the Reviewer. Unfortunately, we did not find more recent studies, conducted with the Siemens software tool in the literature. PMID: 33423941 by Mean et al is now referenced in our revised manuscript, as suggested. Indeed, this study demonstrated not only low inter- and intra-observer variabilities but also good inter-scan reproducibility for the assessment of total and non-calcified plaque volumes in 20 patients who underwent repeated CCTA within 2 weeks. |
Reviewer 2 Report
Comments and Suggestions for Authors
I reviewed with interest the manuscript by Loris Weichsel et al, “Comparison of two Contemporary Quantitative Atherosclerotic Plaque Assessment Tools for Coronary Computed Tomography Angiography: Single-Center Analysis and Multi-Center Patient Cohort Validation.” In this article, the authors showed that the use of the automated CT-guided #2 instrument significantly improved the reproducibility of measuring noncalcified, calcified, and total plaque volumes in less time compared with the standard instrument.
The authors' data are convincing and have undoubted practical significance. However, during the review, I had comments and questions to which I would like to receive answers from the authors.
1. In the Statistical Analysis section, the authors note that they used the Kruskal-Wallis test and the Wilcoxon test when analyzing the results. However, in the text of the article it is not clear what results these tests were used to analyze. Clarification needed.
2. Table 2 does not report statistical differences between groups.
3. There is also no evidence of differences between groups in Suppl Table 2.
4. There is no explanation of the abbreviations used in tables 1-3, this needs to be added.
5. Figures 1-2 are small and difficult to view. Needs to be increased in size.
6. In the Discussion section, some of the information is presented in Table 3. I don't think this is a good solution.
7. Links to sources 22-23 are given in Table 3, but are not discussed in any way in the text of the article. However, these studies reported low Inter-observer variability and Intra-observer variability. Therefore, it would be interesting to compare the results of these studies with the data of the peer-reviewed article.
Comments on the Quality of English LanguageNo comments
Author Response
Thank you very much for taking the time to review this manuscript. Please find the detailed responses below and the corresponding revisions/corrections highlighted/in track changes in the re-submitted files.
|
|
Reviewer 2 |
Reply |
|
Statistical Analysis section |
In the Statistical Analysis section, the authors note that they used the Kruskal-Wallis test and the Wilcoxon test when analyzing the results. However, in the text of the article it is not clear what results these tests were used to analyze. Clarification needed. |
We thank the Reviewer for this comment. Indeed, we used the Mann-Whitney test for the comparison of independent (demographic values, such as age, BMI) and the Wilcoxon test for the comparison of paired variables (time-spent required for the analysis of the same data sets using different software tools). All the continuous variables were non-normally distributed in our manuscript. We clarified this issue in our revised statistical Method section. |
|
Table 2 |
Table 2 does not report statistical differences between groups. |
We thank you for this comment. Since Table 2 reports on variability measures partially conducted in different cohort, we believe that reporting on statistical differences would not be so meaningful in this context. However, to comply with the reviewer’s suggestion, we now report on the statistical differences between inter- and intra-observer variabilities for tool #1 versus tool#2. This comparison is now shown in our revised Figures 1-2 and is also mentioned accordingly in our revised Results section. |
|
Suppl Table 2 |
There is also no evidence of differences between groups in Suppl Table 2. |
We thank you for this suggestion. Unfortunately, the groups in the suppl. Table 2 are largely heterogenous and originate from different cohorts so that a comparison between the different groups would not be very meaningful. In addition, it is well known that radiation exposure is higher with retrospectively gated, compared to prospectively triggered protocols, so that this issue in terms of a direct comparison was not within the scope of our article. |
|
tables 1-3 |
There is no explanation of the abbreviations used in tables 1-3, this needs to be added. |
We thank you for the suggestion. The Abbreviations have now been inserted as suggested. |
|
Figures 1-2 |
Figures 1-2 are small and difficult to view. Needs to be increased in size. |
We thank you again for this suggestion. We now provided high quality PPT and Tiff images of our revised Figures 1 and 2, as suggested. We hope that the resultant image quality is now sufficient. |
|
Discussion section |
In the Discussion section, some of the information is presented in Table 3. I don't think this is a good solution. |
We thank you for the suggestion. The information contained in Table 3 has now been presented in the discussion section. |
|
sources 22-23 |
Links to sources 22-23 are given in Table 3, but are not discussed in any way in the text of the article. However, these studies reported low Inter-observer variability and Intra-observer variability. Therefore, it would be interesting to compare the results of these studies with the data of the peer-reviewed article. |
We thank you for the suggestion. The sources 22-22 have now been inserted in the discussion section. |